# Nano-Topographically Guided, Biomineralized, 3D-Printed Polycaprolactone Scaffolds with Urine-Derived Stem Cells for Promoting Bone Regeneration

**DOI:** 10.3390/pharmaceutics16020204

**Published:** 2024-01-31

**Authors:** Fei Xing, Hui-Yuan Shen, Man Zhe, Kai Jiang, Jun Lei, Zhou Xiang, Ming Liu, Jia-Zhuang Xu, Zhong-Ming Li

**Affiliations:** 1Department of Orthopedic Surgery, Orthopedic Research Institute, Laboratory of Stem Cell and Tissue Engineering, State Key Laboratory of Biotherapy, West China Hospital, Sichuan University, Chengdu 610041, China; xingfeihuaxi@163.com (F.X.); xiangzhoui5@hotmail.com (Z.X.); 2College of Polymer Science and Engineering and State Key Laboratory of Polymer Materials Engineering, Sichuan University, Chengdu 610065, China; shyhuiyuanshen@163.com (H.-Y.S.); jiangkaijkcd@163.com (K.J.); leijun@scu.edu.cn (J.L.); zmli@scu.edu.cn (Z.-M.L.); 3Animal Experiment Center, West China Hospital, Sichuan University, Chengdu 610041, China; zheman@wchscu.cn

**Keywords:** 3D printing, nano-topography, coating, hydroxyapatite, polycaprolactone

## Abstract

Currently, biomineralization is widely used as a surface modification approach to obtain ideal material surfaces with complex hierarchical nanostructures, morphologies, unique biological functions, and categorized organizations. The fabrication of biomineralized coating for the surfaces of scaffolds, especially synthetic polymer scaffolds, can alter surface characteristics, provide a favorable microenvironment, release various bioactive substances, regulate the cellular behaviors of osteoblasts, and promote bone regeneration after implantation. However, the biomineralized coating fabricated by immersion in a simulated body fluid has the disadvantages of non-uniformity, instability, and limited capacity to act as an effective reservoir of bioactive ions for bone regeneration. In this study, in order to promote the osteoinductivity of 3D-printed PCL scaffolds, we optimized the surface biomineralization procedure by nano-topographical guidance. Compared with biomineralized coating constructed by the conventional method, the nano-topographically guided biomineralized coating possessed more mineral substances and firmly existed on the surface of scaffolds. Additionally, nano-topographically guided biomineralized coating possessed better protein adsorption and ion release capacities. To this end, the present work also demonstrated that nano-topographically guided biomineralized coating on the surface of 3D-printed PCL scaffolds can regulate the cellular behaviors of USCs, guide the osteogenic differentiation of USCs, and provide a biomimetic microenvironment for bone regeneration.

## 1. Introduction

In nature, the unique biological functions of various tissues are based on their complex hierarchical nanostructures after long-term evolution and natural selection [1]. Among the various tissues of the human body, dense bone tissue possessing plate-like mineral nanostructures endows bone tissue with excellent mechanical properties that support the various movements of the body in daily life [2]. Bone tissue is a hard and mineralized connective tissue composed of biogenic hierarchical biominerals that exhibit distinct topographies and functions [3]. Biominerals in bone tissue are generated by a biochemical process in body fluids or interstitial fluids, which is called biomineralization [4]. Biominerals inside a living organism consist of various inorganic substances, such as phosphates, carbonates, sulfates, and halides, and emerge as kinds of crystalline, paracrystalline, and amorphous phases [5]. As the hardest substances in vertebrates, the growth and development of bone tissue rely on a self-assembly dynamic biomineralization process. Biomineralization plays a pivotal role in the process of endochondral ossification, facilitating the gradual replacement of cartilage with bone, which is guided by a series of interactive activities of bone tissue and microenvironments. After biomineralization, complex hierarchical nanostructures, morphologies, and unique biological functions are formed in bone tissue [6].

Biomineralization is a complex, multi-level process that deposits inorganic mineralized nanomaterials onto a surface template in a self-assembly manner under mild conditions [7]. The biomineralization process can be adjusted by altering organic templates, inorganic mineralized agents, and external factors, such as pH, time, and temperature [8]. In addition, various nanostructures on the surface of organic templates, such as nanotubes and nanoarrays, can also regulate the biomineralization process. Recently, more and more researchers have utilized biomimetic mineralization as a modification approach to obtain ideal material surfaces with complex nanostructures, unique biological functions, and categorized organizations [9,10,11]. Most biomimetic mineralization approaches are conducted in aqueous environments with controllable conditions. As a calcium phosphate crystal derivative, hydroxyapatite (HA) is the main inorganic substance existing in natural bone tissue and possesses unique crystallographic nanostructures, hierarchically arranged between the collagen fibrils [12]. In vitro, due to similar components to native bone tissue, HA has been widely used as a calcium–phosphate (CaP)-based bone substitute for bone regeneration. After implantation in vivo, the chemical properties of HA are stable. In addition, under physiological conditions, HA has the ability to regulate various cellular behaviors of osteoblasts, such as proliferation, migration, osteogenic differentiation, and cell adhesion [13]. Due to its excellent bioactivity and biocompatibility, HA has gradually become one of the most prominent biomaterials in the fabrication of various bone substitutes. Nevertheless, the direct application of bulk HA as bone implants is limited due to its weak mechanical properties. Researchers have utilized many approaches to enhance the mechanical properties of bulk HA, such as altering the sintering temperature [14]. Simultaneously, applying HA as an additive material coating onto scaffolds, such as metal implants and polymer implants, has also attracted attention to the practical applications of HA [15,16,17].

Due to its excellent biocompatibility, high customizability, good tensile strength, high resistance to degradation, and high stability under physiological conditions, synthetic polymers, including polycaprolactone (PCL), polylactic-co-glycolic acid (PLGA), poly(l-lactic acid) (PLLA), poly(lactic acid) (PLA), and poly(glycolic acid) (PGA), have been widely used as bone substitutes [18,19]. However, the surface of the synthetic polymer scaffolds is bioinert and has the disadvantages of low wettability and weak cell adhesion and thus cannot provide favorable microenvironments for osteogenesis in vivo. Therefore, the biomineralization of HA onto surfaces of synthetic polymer scaffolds can form a bioactive surface with higher wettability and better osteoinductivity, which provides a more favorable microenvironment for osteoblast adhesion, proliferation, and differentiation [20,21]. Currently, soaking in a simulated body fluid (SBF) that contains similar ion levels to those of human body fluid is one of the most common biomineralization approaches and has the advantages of being a simple procedure and having high efficiency [22]. However, biomineralized coating fabricated by immersion in simulated body fluid is uneven and unstable. Researchers can alter the ion concentrations of SBF to regulate biomimetic calcium–phosphate coatings during the biomineralization process [23,24]. In addition, the surface properties, such as topography, charge, and hydrophilicity, greatly affect the surface mineralization of HA during SBF soaking. In recent years, many researchers have altered the surface topography of implants, including nanotubes, nanopits, nanofibers, nanocolloids, nanopillars, nanogrooves, nanodots, and random roughness, to regulate bone formation in vivo [25,26,27,28]. In addition, the surface roughness of scaffolds can directly regulate cellular behaviors and biomineralization on the surfaces of scaffolds [29,30,31]. However, few studies focus on biomineralization on the surface of synthetic polymer scaffolds. Moreover, the propagation of HA in the space between collagens fibrils relies on the continuous supplement of calcium and phosphate ions [32]. Achieving the long-term release of bioactive ions from HA coating plays an important role in continuously regulating bone regeneration in vivo after implantation.

Among various cells in bone tissues, osteoblasts and stem cells are the main source of bone regeneration. Stem cells can differentiate into various cells under various physiochemical or biological activations and directly participate in bone regeneration in vivo [33,34]. As carriers of various stem cells, the surface properties of scaffolds, including physicochemical and biological characteristics, can directly regulate cellular behaviors, such as proliferation, survival, migration, adhesion, and differentiation. Hence, fabricating a favorable surface microenvironment in scaffolds to direct stem cell fate is essential for stem cell delivery in bone repair. HA coating on the surface of scaffolds can provide an osteoinductive microenvironment for stem cells due to its native osteoinductivity. After implantation, the calcium ions and phosphate released from the HA coating can effectively guide stem cells to differentiate into osteoblasts or osteocytes. Recently, the existence of urine-derived stem cells (USCs) in urine has been confirmed by many studies [35]. Compared with other kinds of mesenchymal stem cells (MSCs), USCs possess similar capacities of self-proliferation and multi-potential differentiation [36,37]. Additionally, the isolation procedure of USCs is simple, non-invasive, and cost-effective, which makes it a potential alternative seed cell for stem cell therapy. Therefore, the construction of biomineralized coating on the surface of scaffolds to load and manipulate USCs’ cellular fate can be a promising alternative approach.

In the present study, aiming to enhance the surface osteoinductivity of 3D-printed PCL scaffolds, we optimized the surface biomineralization procedure by nano-topographical guidance. The optimized surface biomineralized coating on the surfaces of customized scaffolds achieved the continuous release of calcium and phosphorus ions and regulated various cellular behaviors of USCs, such as osteogenic differentiation. To this end, the in vivo osteogenesis of nano-topographically guided biomineralized 3D-printed PCL scaffolds combined with USCs was also evaluated. This study provided 3D-printed polymer scaffolds with a promising alternative surface biomineralization strategy to regulate stem cell fate for the guidance of subsequent bone regeneration after implantation.

## 2. Materials and Methods

### 2.1. The Preparation of 3D-Printed PCL Scaffolds

In this study, a 3D bioprinter (Esun 600C, Shenzhen Guanghua Weiye Industrial Co., Ltd., Shenzhen, China) with a printing mechanism of melt extrusion deposition was used to fabricate 3D-printed PCL scaffolds. The viscosity-averaged molecular weight of the PCL pellets used as the raw materials for the PCL scaffolds was 6 × 104 g/mol. In addition, the melting temperature of the PCL pellets was 60 °C. The residual moisture from the PCL pellets was removed by drying at 37 °C for 24 h. Then, the PCL pellets were fabricated into PCL filaments by a single-screw extruder. Then, the PCL filaments were melted in the high-temperature cavity for subsequent shaping. A nozzle with a diameter of 0.5 mm extruded the melted PCL onto the platform surface, and the printed fibers were arrayed orthogonally. The PCL printing temperature was set at 70 °C. The moving speed of the printer nozzle was set at 0.5 mm/s. To this end, the customized PCL scaffolds were vacuum-dried for subsequent surface biomineralization.

### 2.2. Nano-Topographically Guided Biomineralization

Nano-topographically guided biomineralization was conducted through immersion in various functionalized solutions. Nano-topographical modification was essential in this procedure of biomineralization, which was carried out via the approach of epitaxial crystallization. Acetic acid (Chengdu Kelong Co., Ltd., Chengdu, China) and distilled water were mixed at a volume ratio of 77%/23%. Then, the nutrition solution for epitaxial crystallization was fabricated by dissolving PCL pellets into the mixed solution at 60 °C for 3 h. The nano-topographical modification was conducted by immersing 3D-printed PCL scaffolds into the nutrition solution for 15 min at 37 °C. Then, the nano-topographical scaffolds were dried at 37 °C for 24 h. The nano-topographical modification was confirmed by scanning electronic microscopy (SEM, Nova NanoSEM450, FEI Co., Ltd., Hillsboro, OR, USA). Furthermore, the thick distribution of the nano-ridge and the periodic distance distribution of the nano-topographical surface were also calculated according to SEM results.

After surface nano-topographical modification, the subsequent biomineralization procedure was carried out by immersing nano-topographical 3D-printed PCL scaffolds in specific biomineralizing solutions, which were supersaturated four-fold concentrations of simulated body fluids (SBFs). The biomineralizing solutions were prepared by sequentially dissolving 15.95 g of NaCl (Shanghai Macklin Co., Ltd., Shanghai, China), 0.7 g of NaHCO_3_ (Shanghai Macklin Co., Ltd., China), 0.45 g of KCl (Shanghai Macklin Co., Ltd., Shanghai, China), 0.46 g of K_2_HPO_4_·H_2_O (Shanghai Macklin Co., Ltd., Shanghai, China), 0.56 g of CaCl_2_ (Shanghai Macklin Co., Ltd., Shanghai, China), 0.61 g of MgCl_2_·6H_2_O (Shanghai Macklin Co., Ltd., Shanghai, China), and 0.14 g of Na_2_SO_4_ (Shanghai Macklin Co., Ltd., Shanghai, China) into 500 mL of deionized water. Then, the Tris-HCl solution was used as a buffer to adjust the pH value of the biomineralizing solutions to 7.4. The nano-topographically guided biomineralized scaffolds were prepared by immersing nano-topographical scaffolds in biomineralizing solutions at 37 °C for seven days. The biomineralizing solutions were changed every 24 h during the biomineralized coating formation on the surface of the scaffolds. Simultaneously, as a control group, bare 3D-printed PCL scaffolds without nano-topographical modification were also immersed into biomineralized solutions at 37 °C for seven days to form biomineralized scaffolds. After this procedure of biomineralization, biomineralized scaffolds were dried at 37 °C for 24 h. To this end, the bare 3D-printed PCL scaffolds, nano-topographical 3D-printed PCL scaffolds, biomineralized 3D-printed PCL scaffolds, and nano-topographically guided biomineralized 3D-printed PCL scaffolds were named BS, NS, MS, and NMS, respectively. All scaffolds were sterilized by ethylene oxide at a low temperature for subsequent cell and animal experiments.

### 2.3. Characteristics of Scaffolds

During the processes of nano-topographically guided biomineralization and conventional biomineralization, the surface morphology of BS, NS, MS, and NMS at nano and micro levels was observed by SEM (Nova NanoSEM450, FEI, Charlottesville, VA, USA). In addition, the water contact angle of BS, NS, MS, and NMS was also investigated. During the procedure of biomineralization, the surface chemistry of BS, NS, MS, and NMS was evaluated by Fourier transform infrared spectroscopy (FTIR, Nicolet 6700, Thermal Scientific Co., Ltd., Waltham, MA, USA). In addition, the wavenumber range of FTIR was set at 700–4000 cm^−1^. The crystallographic structure was evaluated by X-ray diffraction (XRD, Geigerflex Co., Ltd., Rigaku, Japan). In addition, the scan range was set at 5–60°. The surface elements were assessed by X-ray photoelectron spectroscopy (XPS, XSAM800, Shimadzu-Kratos Ltd., Tokyo, Japan). Moreover, the C (1s), O (1s), Ca (2p), and P (2p) core-level spectra of the XPS results were also compared. A universal testing machine (Instron 5976, Thermal Scientific Co., Ltd., Waltham, MA, USA) was used to assess the effects of biomineralized coating on the mechanical properties. Furthermore, stress–strain and force–displacement curves were calculated.

The surface characteristics of the biomineralized coating of MS and NMS were further investigated to compare the approaches of nano-topographically guided biomineralization and conventional biomineralization. Calcium (Ca) and phosphorus (P) were the two main components of the biomineralized coating. The Ca and P ion distribution on the surface of MS and NMS was investigated by elemental energy dispersive spectrum (EDS). The Ca ions are marked green, while the P ions are marked red. The thermal stability of MS and NMS was measured by thermogravimetric analysis (TGA). In addition, the TG–temperature curves of MS and NMS were calculated. The release of Ca and P ions is an important aspect in investigating the characteristics of biomineralized coating. The bioactive ions released from MS and NMS at different time points were evaluated by inductively coupled plasma atomic emission spectroscopy (ICP-AES; Vista AX, Thermal Scientific Co., Ltd., Waltham, MA, USA). The protein adsorption capacity of MS and NMS was investigated by the protein estimation kit (Beyotime, BCA Protein Assay Kit, Shanghai, China). In order to investigate the adhesion of the biomineralized coating of scaffolds, the scaffolds were immersed in PBS and placed on the horizontal shaker. Then, the scaffolds were immersed in an alizarin red solution for 30 min.

### 2.4. Isolation of USCs

According to our previous studies, the USCs were isolated and obtained by the method of two times centrifugations [35,36]. After obtaining informed consent from all urine donors, 200–250 mL of fresh urine from healthy adults was collected into a sterile glass beaker. Then, the urine was separated into several 50 mL centrifuge tubes. Then, the centrifuged speed of the centrifuge tubes was set at 1700 rpm and the centrifuged time was set to 15 min. Then, the tubes were centrifuged at a centrifuge speed of 1700 rpm for 15 min. The pellets of the solution were resuspended with a specific culture medium for USCs, which is fabricated by mixing embryo fibroblast medium (EFM) (Thermo Fisher Scientific Inc., Fremont, CA, USA) and keratinocyte serum-free medium (KSFM) (Thermo Fisher Scientific Inc., Fremont, CA, USA) at a volume ratio of 1:1. Then, the suspension of USCs was seeded in cell culture plates.

### 2.5. Cellular Behaviors of USCs Co-Cultured with MS and NMS

In order to investigate the cellular behaviors directed by nano-topographically guided biomineralized coating, the USCs were co-cultured with MS and NMS. The cellular behaviors include proliferation, cell morphology, cell viability, migration, and invasion. Co-culturing USCs with scaffolds was performed in Transwell plates (Corning. Inc., New York, NY, USA). The USCs were seeded onto the lower chamber, while the scaffolds were placed in the upper chamber. Cell Counting Kit-8 (CCK-8, Beyotime Biotech. Inc., Shanghai, China) was applied to quantitatively evaluate the proliferation of USCs co-cultured with MS and NMS at different time points. Briefly, after discarding the medium, the CCK8 working solution was used to incubate the USCs for 1 h. Then, the absorption values of the CCK8 working solution were texted. The cytotoxicity of the nano-topographically guided biomineralized coating was investigated by live/dead staining (Beyotime Biotech. Inc., Shanghai, China) of the USCs co-cultured with MS and NMS on day 1, day 4, and day 7. The live/dead staining solution contained two fluorescent dyes to label live and dead cells separately. The live USCs were stained green and the dead USCs were stained red. After incubation with a working solution for 5 min, the samples were analyzed by fluorescent microscopy. In addition, the cell viability of USCs was also calculated according to the live/dead staining results. Cytoskeleton changes can affect various cellular behaviors. Phalloidin/DAPI staining (Beyotime Biotech. Inc., Shanghai, China) was used to observe the cytoskeleton changes of USCs co-cultured with MS and NMS. Briefly, after discarding the supernatant, the USCs were fixed in PBS containing 4% formaldehyde for 1 h. Then, the USCs were immersed in a 0.1% Triton X-100 solution for 30 min at room temperature. Then, the USCs were treated with a phalloidin staining working solution for 1 h in the dark. After being treated with a DAPI working solution, the cytoskeleton changes were observed by fluorescent microscopy. In addition, the cell area of USCs was calculated according to the cytoskeleton staining results.

The cell migration and invasion of USCs co-cultured with MS and NMS were investigated by the scratch–wound assay and transwell invasion assay. The USCs were seeded onto a microwell plate. When the USCs covered the bottom of the plate, the cell layer was scraped in a straight line using a 1 mm pipette tip. Then, the microscope was used to image the same pot at 0 h, 6 h, 12 h, and 24 h. Different from the above co-culture approach, the USCs were seeded on the top chamber of the transwell plates and the scaffolds were placed on the lower chamber during the transwell invasion assay. After incubation for 24 h, USCs that migrated to the lower surface of the top chamber membrane were fixed with 4% formaldehyde and stained with DAPI. The number of migrated USCs was calculated. In addition, in order to observe cellular behaviors on the surface of MS and NMS, the USCs were also directly seeded onto the surface of MS and NMS. Live/dead staining and phalloidin/DAPI staining were used to investigate the proliferation and cytoskeleton changes of USCs when directly co-cultured on the surface of scaffolds. In addition, the cell viability and cell area of USCs on the biomineralized coating were also calculated.

### 2.6. The Osteogenesis of USCs Co-Cultured with MS and NMS

The osteogenesis of USCs co-cultured with MS and NMS at different time points was conducted in transwell plates. The USCs were seeded onto the lower chamber while the scaffolds were placed on the top chamber. Alkaline phosphatase (ALP) staining was used to investigate the early osteogenesis of USCs regulated by nano-topographically guided biomineralized coating. Briefly, after co-culturing USCs with scaffolds, the medium was discarded and fixed with 10% formaldehyde for 15 min. Then, the USCs were incubated in an ALP buffer for 15 min. The BCIP/NBT color development substrate solution was used to stain USCs. In addition, the positive areas of ALP staining results were also calculated. Alizarin red staining (ARS) is a common approach for investigating the mineralization of various osteoblasts. On day 14 and day 28, the medium in the transwell plate was discarded. Then, the USCs were immersed in an alizarin red working solution for 30 min. Moreover, the positive area of ARS was assessed to quantitatively analyze the USCs’ mineralization regulated by nano-topographically guided biomineralized coating. After co-culturing with MS and NMS for 7 and 21 days, the osteogenesis-related gene expression of the USCs, including collagen type 1 alpha 1 (COL1A1), alkaline phosphatase (ALP), osteocalcin (OCN), and runt-related transcription factor 2 (RUNX2), was investigated by using RT-qPCR. In addition, GAPDH was used as a reference gene. Appendix A reports the primer sequences of the target genes. Briefly, the TRIzol solution was applied to extract the total RNA of cells. Then, RNA was reverse-transcribed into cDNA by using the QuantiTect Reverse Transcription Kit. Then, the amplification of the specific transcripts on the real-time fluorescence quantitative instrument was completed on a LightCycler 96 system by utilizing the SYBR Green Mix Kit. The condition was set at 94 °C for 5 min followed by 40 cycles of 94 °C for 15 s, 55 °C for 30 s, and 72 °C for 30 s. The 2-DDCq method was utilized to evaluate the relative levels of osteogenic gene expression. In addition, the USCs were seeded onto the surface of MS and NMS to assess the direct effect of scaffolds on cellular behaviors. In addition, live/dead staining and phalloidin staining were used to investigate the proliferation and cytoskeleton changes of USCs after being seeded onto the surface of scaffolds. The cell viability and cell area of USCs on the surface of MS and NMS were also calculated. Moreover, the osteogenesis-related gene expression of USCs on the surface of MS and NMS, including COL1A1, ALP, OCN, and RUNX2, was evaluated by using RT-qPCR.

### 2.7. In Vivo Bone Regeneration

All animal procedures were performed in accordance with the Guidelines for Care and Use of Laboratory Animals of West China Hospital, Sichuan University, and experiments were approved by the Animal Ethics Committee of West China Hospital, Sichuan University. The rabbit cranial bone defect models were used as in vivo osteogenesis models to evaluate the bone regeneration mediated by NMS loaded with USCs. Twenty New Zealand white rabbits (body weight: 2.5–3.0 kg; gender: male) were randomly separated into five groups: NMS loaded with USCs (NMS/USCs, n = 4), MS loaded with USCs (MS/USCs, n = 4), NMS (n = 4), MS (n = 4), and a control (n = 4). In groups of NMS/USCs and MS/USCs, the USCs were seeded onto the surface of NMS and MS and then implanted into cranial bone defects. The NMS and MS were implanted in groups of NMS and MS. In the control group, no scaffolds were implanted.

The procedure for the in vivo osteogenesis model was as follows. Briefly, after general anesthesia, the surgeon performed an incision with a longitudinal length of 4 cm. Then, the subcutaneous tissue under the skin was cut layer by layer. After the removal of the periosteum of the cranial, the cranial was fully exposed. After the construction of two circular defects with a diameter of 10 mm, the scaffolds were implanted. The information on animal grouping was reported above. After the implantation of scaffolds, the wounds were closed layer by layer with 4-0 sutures. After surgery, intramuscular penicillin was administered for 3 days at a dose of 100,000 U/day to prevent the occurrence of infection.

All animals were sacrificed by overdose of anesthesia at 8 and 16 weeks after implantation. The micro-computed tomography (Micro-CT, Trifoil Imaging Inc., Chatsworth, CA, USA) was utilized to observe newborn bone formation. In addition, the BV/TV and bone density of new bone were calculated. After imaging evaluation, the cranial samples were fixed with 10% formaldehyde and then immersed in a decalcification solution of EDTA for 8 weeks. Histological staining, including HE and Masson staining, was used to assess the new bone formation in the bone defect site. Moreover, new bone areas in the middle of the bone defect were also calculated according to the histological results.

### 2.8. Statistical Analysis

All experimental data analyses were performed by SPSS (Version 25). The statistical analysis of significant differences between the study groups was conducted by using the Mann–Whitney U-test or Student’s *t*-test. Furthermore, *p*-values of <0.05 were interpreted as statistically significant.

## 3. Results and Discussion

### 3.1. The Fabrication of Nano-Topographically Guided Biomineralized 3D-Printed PCL Scaffolds

Currently, due to their good biocompatibility and abundant resources, various polymer scaffolds have been used as orthopedic implants such as joint prostheses, bone screws, nails, and plates. However, the bioinert and poor osteoinductive surface of polymer implants cannot chemically bind to surrounding bone tissue, resulting in limiting their further application for orthopedic implants [38]. Thus, the formation of a natural bone-like hydroxyapatite coating layer on polymer scaffolds plays an important role in enhancing osteointegration after implantation. In addition, few studies focus on fabricating biomineralized coating on the surface of 3D-printed polymer scaffolds. In our study, we report a simple biomineralization approach for 3D-printed polymer scaffolds via surface nano-topographical modification. Figure 1A shows the fabrication procedure for the 3D-printed PCL scaffolds. The procedures for the nano-topographically guided mineralization and conventional mineralization of the 3D-printed polymer scaffolds are shown in Figure 1B,C. Conventional mineralization was conducted by immersing the scaffolds into mineralized solutions. As shown in Figure 1D, the general views of BS, NS, MS, and NMS exhibited a similar appearance of white mesh. In addition, according to the SEM results, no significant differences were found among BS, NS, MS, or NMS when magnified to the micron level. When magnified to the nanoscale, BS, NS, MS, and NMS possessed totally different surface nano-topographies. The surface of BS was flat and smooth, while the NS exhibited a nano ridge-like surface. Altering surface properties, such as topography, roughness, and grain structure, has proven to be an effective approach to enhance the osteointegration of implants [39,40]. Moreover, our previous study demonstrated that the topography surface of polymer sheets fabricated by surface epitaxial crystallization could provide sufficient crystallization sites to enhance the in vitro biomineralization and osteogenic differentiation of preosteoblasts [41,42]. After conventional biomineralization, some mineralized crystals with irregular shapes could be observed on the MS surface. After nano-topographically guided biomineralization, a large amount of mineralized crystals with clustered shapes formed on the surface of NMS. In terms of section views of the scaffolds, some scattered mineralized crystals could be observed on the MS surface, while there was still a large amount of mineralized crystal formed on the inner fibers. In addition, compared with MS, the mineralized crystals of NMS almost fully covered the PCL fibers.

### 3.2. Characteristics of Nano-Topographically Guided Biomineralized 3D-Printed PCL Scaffolds

The thick distribution of the nano-ridge and periodic distance distribution are shown in Figure 2A,B. Based on the SEM results, the average thickness of the nano-ridge on the surface of NS was 92.04 ± 13.69, while the average periodic distance of the nano-ridge on the surface of NS was 725.639 ± 103.93 nm. In addition to adjusting the ion concentrations of the SBF solution, changing the surface properties of scaffolds can also affect the biomineralization process [23,24]. Compared with the flat surface of NS, the nano-ridge increased the surface area as well as the roughness and provided more binding sites for subsequent biomineralization. The water contact angle of the surface of bone substitutes is important in regulating the various cellular behaviors of stem cells. As shown in Figure 2C, the water contact angle of BS, NS, MS, and NMS was 110.21 ± 3.88°, 105.71 ± 4.27°, 79.85 ± 5.12°, and 53.62 ± 5.33°. After biomineralization, the biomineralized coating gave a lower surface water contact angle for MS and NMS due to the deposition of mineralized substances. In addition, the water contact angle of NMS was significantly lower than that of MS, which indicated the greater existence of mineralized substances deposited onto the scaffolds. In the field of bone repair, the surface charge of biological scaffolds plays an important role in regulating various cell behaviors [43,44]. The Zeta potential of BS, NS, MS, and NMS was −49.44 ± 3.75 mV, 58.35 ± 4.04 mV, 43.37 ± 3.15 mV, and −18.71 ± 3.57 mV at a pH of 7.4, respectively. Moreover, as shown in Figure 2D,E, FTIR and XRD were applied to observe the chemical composition and crystal structure of the surface of BS, NS, MS, and NMS. The FTIR results demonstrated that BS and NS shared the same strong peaks at 1750 cm^−1^ and 1084 cm^−1^ (C=O and C-O-C), which indicated that the nano-topographical surface fabricated by epitaxial crystallization possessed a similar composition to that of the base material. After biomineralization, the surface FTIR of NMS and MS showed characteristic peaks of HA at 1046 cm^−1^, which confirmed the existence of biomineralized coating. In addition, compared with MS, the characteristic peaks of the HA of NMS were more obvious and even covered the characteristic peaks of PCL, which indicated a greater existence of HA on the scaffolds’ surface. XRD is a common technique to evaluate the type and content of crystal phases. XRD demonstrated that NS and BS shared a similar XRD curve, which confirmed that nano-topographical modification does not introduce new substances to the scaffold surface. Furthermore, the XRD results also revealed the typical peaks of the HA of NMS and MS. In addition, compared with MS, NMS possessed higher typical peaks of HA at 31.9°, which also demonstrated that more HA had deposited onto the scaffold surface. Figure 2F shows the XPS results of BS, NS, MS, and NMS. Figure 2G,J show the C (1s), O (1s), Ca (2p), and P (2p) core-level spectra of BS, NS, MS, and NMS, which demonstrated the obvious existence of Ca and P substances on the surface of NMS. The two kinds of biomineralized approaches could deposit HA onto the surface of PCL, while nano-graphically guided biomineralization could deposit more mineralized substances onto polymer scaffolds. In addition, Figure 2K,L show that BS, NS, MS, and NMS shared similar stress–strain curves and force–displacement curves. The mechanical properties showed that nano-topographical medication and mineralization did not change the mechanical properties of the scaffolds. The FTIR, XPS, and XRD results demonstrated the existence of biomineralized coating on the surface of MS and NMS, which is consistent with the above SEM results.

In natural bone tissue, as the main inorganic component, HA can regulate various osteoblasts and reinforce the collagen fibrils as the basic building blocks. Fabricating biomineralized coating on polymer scaffolds plays an important role in structural and functional bionics, leading to scaffolds that are close to natural bone in composition and structure. The surface nano-topography of scaffolds can effectively affect the biomineralized coating. Thus, the development of biomineralization based on surface topology can enable the regulation of the procedure of biomineralization without introducing new chemical components into scaffolds. In our study, two kinds of biomineralization coating were further compared. The SEM-EDS elemental mapping of NMS and MS is shown in Figure 2M. The green and red dots represent the existence of Ca and P ions, respectively. The Ca and P ions were evenly distributed on the biomineralized coating of MS and NMS. Compared with MS, more Ca and P ions were deposited on the surface of the scaffolds, which indicated that a nano-topographical surface can effectively enhance biomineralization. Thermogravimetric (TG) analysis is an analytical technique used to investigate the weight change of biomaterials, such as polymers, under controlled temperatures. Figure 3N shows the TG curves of MS and NMS. With the temperature heated up to over 300 °C, the weight of MS and NMS decreased dramatically. With the temperature heated up to over 500 °C, the weight of MS and NMS remained stable. Due to the native TG differences between polymer and biomineralized coating, most of the remaining substances, when heated up to over 500 °C, were inorganic biomineralization substances. The TG results show that the remaining substance weight of NMS was higher than that of MS, which indicated that the method of nano-topographically guided biomineralization can deposit more inorganic biomineralization substances than conventional mineralization. Protein adsorption occurring on the surface of scaffolds is an essential mediator of various biological responses. In addition, the interaction between the scaffold and cells in the human body after implantation is achieved by protein adsorption [45]. As shown in Figure 2O, the protein adsorption of NMS was higher than that of MS. The higher protein adsorption capacity of the nano-topographically guided biomineralized coating might enhance the polymer scaffolds’ surface bioactivity for bone regeneration. The osteoinductivity of biomineralized coating is achieved by releasing bioactive Ca and P ions, which are involved in various complex biological processes, such as bone regeneration and bone remodeling. In addition, the Ca and P ions released from biomineralized coating on the surface of scaffolds can directly take part in directing the fate of stem cells. Figure 2P,Q show the Ca and P ions released from MS and NMS, respectively. At day 7, the level of Ca and P ions released from MS was higher than that from NMS, which could be attributed to the adhesion of the biomineralized coating to the scaffold. After day 7, the Ca and P ion levels of the NMS group continued to rise, and the level at day 28 was almost four times the level at day 7. As for the MS group, the Ca and P ion levels increased in a small range. The Ca and P ion levels of NMS were significantly higher than those of MS on day 14, day 21, and day 28. The long-term efficient release of Ca and P ions from biomineralized coating is conducive to the long-term regulation of stem cells’ biological activities and in vivo bone regeneration. The whole period of bone regeneration goes on for several months [46] and, thus, an ideal biomineralized coating needs to possess a high adhesion capacity to scaffolds, which can continuously regulate the process of osteogenesis after implantation. Figure 3F shows the ALP staining of MS and NMS after horizontal shaking in PBS. After shaking, the biomineralized coating on the surface of MS gradually fell off. By day 7, the biomineralized coating on the surface of the MS almost completely fell off. The biomineralized coating constructed by nano-topographical guidance could stably exist on the surface of NMS after shaking.

### 3.3. Cellular Behaviors of USCs Co-Cultured with MS and NMS

USCs were co-cultured with MS and NMS in transwell plates to investigate the cellular behaviors affected by scaffolds. The biomineralized coating on the surface of 3D-printed PCL scaffolds is able to promote bone regeneration by regulating cellular behaviors and the fate of stem cells. Among these cellular behaviors, as the basic functions of stem cells, the cell proliferation ability has been widely used to investigate the cytocompatibility and cytotoxicity of scaffolds. Maintaining normal cell morphology is essential for maintaining the biological viability of stem cells. Figure 3A shows the cytoskeleton staining of USCs co-cultured with scaffolds. Compared with the control group, USCs co-cultured with MS and NMS possessed normal cell morphology. In addition, no significant differences were found in terms of the cell area calculated according to the control, MS, and NMS. (Figure 3G) The CCK-8 results of the USCs co-cultured with MS and NMS at different time points are shown in Figure 3E. On day 3, the OD value of MS was significantly higher than those of the control and NMS. Furthermore, no significant differences were found between the groups control, MS, and NMS, which indicated the good cytocompatibility of MS and NMS. In order to investigate the visible cell proliferation of USCs co-cultured with scaffolds, live/dead staining was also conducted. With the extension of the culture time, each group of cells exhibited a stable cell proliferation capacity. Compared with the control group, the USCs co-cultured with MS and NMS showed a similar cell proliferation capacity (Figure 3B). In addition, the cell viability of USCs when co-cultured with scaffolds was calculated according to the live/dead staining results (Figure 4F). No significant differences were observed among the groups control, MS, and NMS, which also demonstrated the good cytocompatibility of MS and NMS. Cellular morphology changes contribute to various cellular functions such as cell division, signaling, and differentiation [47,48]. Cell migration is an important process involving many biological activities, such as tissue regeneration, development, inflammation, and differentiation [49]. In order to investigate the cell migration and invasion of USCs co-cultured with MS and NMS, scratch–wound and transwell invasion assays were performed. As shown in Figure 3C, the scratch–wound assay results demonstrated that the MS and NMS could enhance the migration of USCs compared with the control group. Additionally, the scratch area of MS and NMS, calculated according to the scratch–wound assay results, was significantly lower than that of the control group at 6 h and 12 h (Figure 3H). The transwell invasion assay results are shown in Figure 3D. No significant differences in cell numbers were found among the groups MS, NMS, and control at 12 h. The invasion cell numbers of MS and NMS were significantly higher than those of the control group at 24 h, which demonstrated that MS and NMS could effectively enhance the invasion of USCs (Figure 3I).

### 3.4. The Osteogenesis of USCs Co-Cultured with MS and NMS

Ideal biomineralized coating on the surface of scaffolds can mimic the microenvironment of natural bone tissue, provide appropriate biochemical and physical signals, guide the osteogenic differentiation of stem cells, and enhance the deposition and mineralization of bone matrix after implantation [50]. Alkaline phosphatase (ALP) is highly expressed in the cells of mineralized tissue and plays an important role in hard tissue formation. In addition, ALP has been considered to be an initial marker during the process of osteogenic differentiation. Figure 4A shows the ALP staining results of USCs co-cultured with scaffolds. The positive area of ALP staining is calculated in Figure 4B. The ALP-positive areas of MS and NMS were significantly higher than that of the control group on day 5 and day 10. In addition, the ALP-positive area of NMS was significantly higher than that of MS on day 10. Alizarin red staining (ARS) was used to evaluate calcium-rich deposits by cells in the culture. As shown in Figure 4C, the ARS results show that more calcium nodules were found in the NMS group than in other groups. The positive area of the ARS results is also calculated in Figure 4D. The ARS-positive area of the NMS group was higher than that of the groups MS and control on day 14 and day 28. In addition, the ARS-positive area of the MS group was higher than that of the control group. Figure 4E shows the osteogenic-related gene expression of USCs co-cultured with scaffolds. As a member of the RUNX family of transcription factors, RUNX2 regulates the process of osteogenic differentiation. The RUNX2 expression of the NMS group was significantly higher than that of the groups MS and control on day 7 and day 21. ALP is produced mainly by bones and the liver. In addition, ALP can take part in hydrolyzing inorganic pyrophosphate and provide inorganic phosphate to modulate the process of biomineralization in bone tissues. The ALP expression of groups NMS and MS was significantly higher than that of the control group. As an early maker of osteogenesis, COL1A1 plays a crucial role in the formation of collagen fiber and the strengthening of bone tissue. The COL1A1 expression of the NMS group was significantly higher than that of the groups MS and control on day 7 and day 21. As a late maker of osteogenesis, osteocalcin is produced by osteoblasts or stem cells. In addition, OCN is a non-collagen protein and plays a vital role in regulating the processes of mineral deposition and bone remodeling. The OCN expression of groups of NMS and MS was significantly higher than that of the control group on day 7 and day 21. Additionally, the OCN expression of the NMS group was significantly higher than that of MS on day 21. Compared with biomineralized coating fabricated by the conventional method, the biomineralized coating fabricated by nano-topographical guidance showed a better regulation capacity in guiding the osteogenic differentiation of USCs.

In our study, USCs were also seeded onto the scaffolds to evaluate the direct effect of two mineralized scaffolds on cellular behaviors. The proliferation and cellular morphology changes of stem cells on the surface of scaffolds play a very important role in osteogenesis in vitro and in vivo. As shown in Figure 5A, the USCs on the surface of MS and NMS exhibited a stable proliferation ability. With the expansion of the culture time, the number of USCs on the scaffolds gradually increased. Compared with the MS group, more USCs could be observed on the surface of NMS. On day 7, USCs on the surface of NMS could fuse together and cover the scaffold fiber. However, as shown in Figure 5B, no significant differences were found in terms of cell viability between the groups MS and NMS. Figure 5C shows the results of the phalloidin staining of USCs seeded onto scaffolds on day 7. The USCs on the surface of MS were scattered while the USCs on the surface of NMS merged. In addition, as shown in Figure 5D, the cell area of the NMS group was significantly higher than that of the MS group on day 4. Figure 5D shows the osteogenesis-related gene expression of USCs on the surface of MS and NMS. The COL1A1, ALP, and RUNX2 expression of the NMS group was significantly higher than that of the MS group, which confirmed the direct contribution to osteogenesis in vitro.

### 3.5. In Vivo Bone Regeneration

Micro-CT was utilized to observe newborn bone formation in the bone defects after implantation. Figure 6A shows the micro-CT results of scaffolds implanted into bone defects. At 8 weeks after implantation, no obvious newborn bone formation was found in the control group. In the MS group, a small amount of newborn bone formation was found at the edge of bone defects. In the NMS group, more newborn bone formation was found at the edge of bone defects. In the MS/USCs group, a small amount of newborn bone formation was found in the central area of bone defects. In the NMS/USCs group, newborn bone formation could be found at the edge and central area of bone defects. At 16 weeks after implantation, only a small amount of newborn bone formation was found at the edge of bone defects in the control group. In the MS group, the newborn bone formation covered part of the bone defects. Compared with the MS group, more newborn bone formation was found in the NMS group. In the MS/USCs group, newborn bone formation occurred inside the scaffolds. In the NMS/USCs group, a large amount of newborn bone fused together in the bone defects. As shown in Figure 6B, BV/TV was also calculated to quantitatively investigate the bone formation occurring at the site of bone defects. The BV/TV of each scaffold group was significantly higher than that of the control group at 8 weeks and 16 weeks after implantation. The BV/TV of the NMS group was significantly higher than that of the MS group. The BV/TV of the NMS/USCs group was significantly higher than that of the other groups. No significant differences were found between the NMS group and the MS/USCs group. Moreover, newborn bone density in the bone defects was also calculated according to the micro-CT results. As shown in Figure 6C, the results show that all groups shared similar newborn bone density, which confirmed the homogeneity of the newborn bone tissue.

After implantation, the biomineralized coating on the surface of polymer scaffolds is able to provide the proper microenvironment for bone regeneration by continuously releasing Ca and P ions [51]. The crystals of calcium and phosphorus compounds in the biomineralized coating can serve as a template for bone matrix formations. In addition, various proteins can adhere to the surface of scaffolds and mediate and regulate the process of bone formation. The HE staining results of the central area of the bone defects at 8 and 16 weeks after implantation are shown in Figure 7. At 8 weeks after implantation, no obvious newborn bone formation was found on the bone defects of the control group. In the MS group, only a small amount of newborn bone formation was found at the edge of the bone defects. In addition, some of the fibrous tissue was distributed around the scaffold fiber. Compared with the MS group, more newborn bone formation was distributed inside the NMS scaffolds. Furthermore, newborn bone formation mainly occurred around scaffold fibers. In the MS/USCs group, many fibrous tissues were found inside the scaffolds and no obvious newborn formation occurred in bone defects. In the NMS/USCs group, newborn bone formation occurred inside the scaffolds. Compared with the NMS group, more newborn bone formation was distributed in the central area of bone defects. At 16 weeks after implantation, only a small amount of newborn bone formation occurred at the edge of bone defects. Compared with the MS groups, more newborn bone formation occurred along the scaffold fiber in NMS. In the MS/USCs group, few newborn bone formations were found inside the scaffolds. Compared with the other groups, a large amount of newborn bone formation was found in the NMS/USCs group. In addition, the newborn bone formed along the scaffold fibers and almost filled the inside of the scaffolds.

The Masson staining results of the central area of bone defects at 8 and 16 weeks after implantation are shown in Figure 8. The blue area represents the mineralized bone of newborn bone formation and the red area represents the osteoid, unmineralized bone tissue [52]. The Masson staining results of all groups were consistent with the HE staining results. According to the Masson staining results, the newborn bone tissue of all groups was calculated to quantitatively investigate the bone regeneration in the central areas of bone defects (Figure 9). The new bone area of each scaffold group was significantly higher than that of the control group at 8 and 16 weeks after implantation. The new bone area of the NMS group was significantly higher than that of the MS group at 8 and 16 weeks after implantation. Interestingly, the new bone area of the NMS group was significantly higher than that of the MS/USCs group at 16 weeks after implantation, which demonstrated that effective biomineralized coating is better than stem cells combined with weak biomineralized coating for the long-term regulation of osteogenesis in vivo. In addition, the new bone area of the NMS/USCs group was significantly higher than that of the other groups at 8 and 16 weeks after implantation, which demonstrated that nano-topographically guided biomineralized coating combined with USCs can exert a good synergistic effect on the induction of osteogenesis in vivo.

## 4. Conclusions

In the current study, we fabricated a biomineralized coating on the surface of 3D-printed PCL scaffolds by nano-topographically guided modifications for bone regeneration. Compared with biomineralized coating constructed by the conventional method, the nano-topographically guided biomineralized coating possessed more mineral substances and more firmly existed on the surface of the scaffolds. In addition, nano-topographically guided biomineralized coating possessed a better protein adsorption capacity and Ca–P ion release capacity. Regarding the cellular behaviors of USCs co-cultured with scaffolds, the nano-topographically guided biomineralized coating could maintain the proliferation capacity of USCs and enhance the migration and invasion of USCs. Regarding the in vitro osteogenesis of USCs co-cultured with scaffolds, compared with biomineralized coating constructed by the conventional method, the nano-topographically guided biomineralized coating could better promote the in vitro mineralization of USCs and the expression of osteogenesis-related genes. After being implanted into bone defect models, compared with conventional biomineralized scaffolds, nano-topographically guided biomineralized scaffolds exhibited a better ability to induce bone regeneration. In addition, the combination of nano-topographically guided biomineralized scaffolds with USCs exhibited a good synergistic effect on the induction of osteogenesis in vivo. The present work demonstrated that nano-topographically guided biomineralized coating on the surface of 3D-printed PCL scaffolds can achieve the long-term release of Ca and P ions, regulate the cellular behaviors of USCs, guide the osteogenic differentiation of USCs, and provide a biomimetic microenvironment for in vivo bone regeneration after implantation. Overall, our study provides 3D-printed polymer scaffolds with a promising alternative surface biomineralization strategy to regulate stem cell fate for the guidance of subsequent bone regeneration after implantation.

## Figures and Tables

**Figure 1 pharmaceutics-16-00204-f001:**
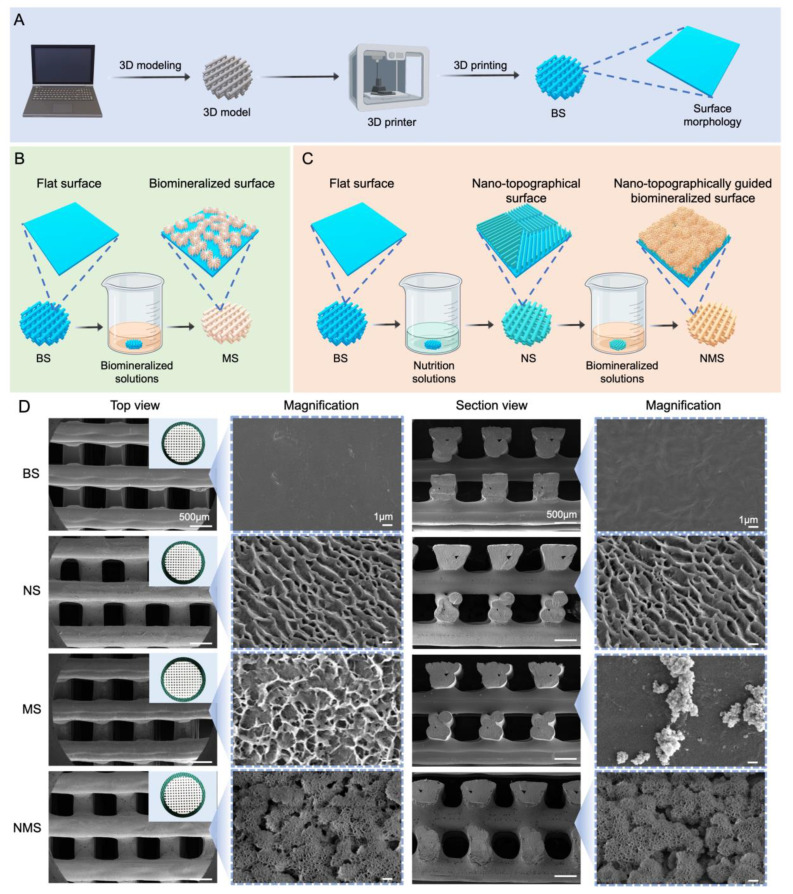
(**A**) The fabrication of 3D-printed PCL scaffolds. (**B**) The procedure for conventional biomineralization. (**C**) The procedure for nano-topographically guided biomineralization. (**D**) General view and SEM results of BS, NS, MS, and NMS, respectively.

**Figure 2 pharmaceutics-16-00204-f002:**
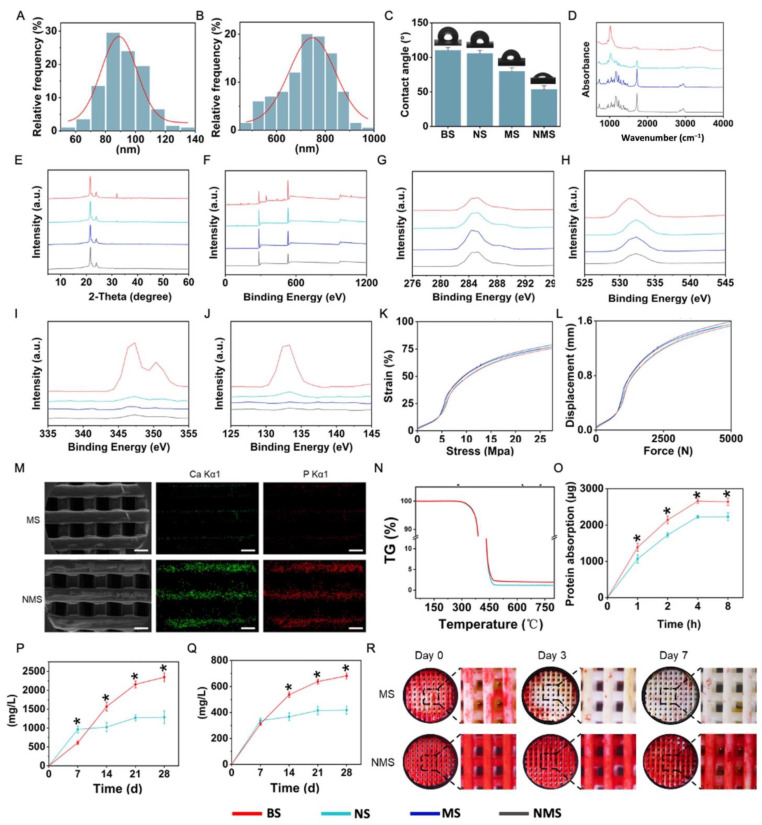
(**A**) The thick distribution of the nano-ridge on the surface of NS. (**B**) The periodic distance distribution on the surface of NS. (**C**) The water contact angle. (**D**) The FTIR results of the scaffolds. (**E**) The XRD results of the scaffolds. (**F**) The XPS results of the scaffolds. (**G**–**J**) The C (1s), O (1s), Ca (2p), and P (2p) core-level spectra of BS, NS, MS, and NMS. (**K**) The stress–strain curves of the scaffolds. (**L**) The force–displacement curves of the scaffolds. (**M**) The SEM-EDS elemental mapping of the scaffolds. Scale bar = 500 μm (**N**) The TG curve of the scaffolds. (**O**) The Ca ions released from the scaffolds. (**P**) The P ions released from the scaffolds. (**Q**) The protein absorption of the scaffolds. (**R**) The ARS results of the scaffolds. * *p* < 0.05.

**Figure 3 pharmaceutics-16-00204-f003:**
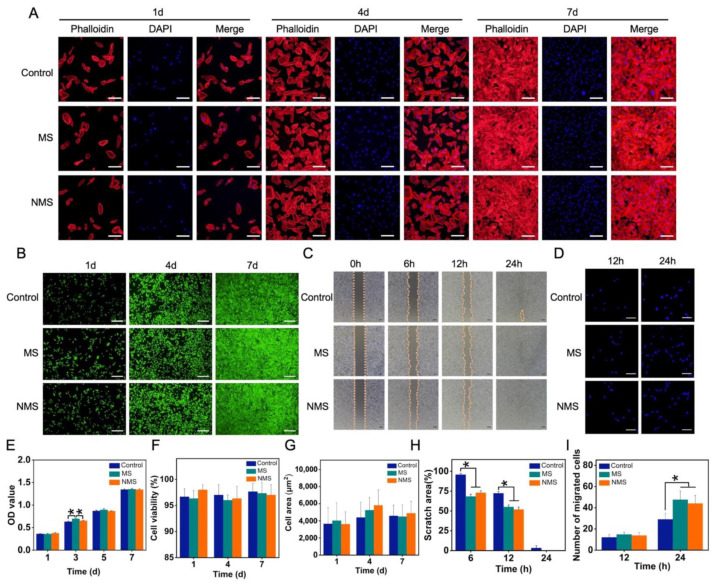
(**A**) Phalloidin/DAPI staining of USCs co-cultured with scaffolds. (**B**) Live/dead staining of USCs co-cultured with scaffolds. (**C**) The scratch–wound assay of USCs co-cultured with scaffolds. (**D**) The number of migrated cells. (**E**) The CCK-8 results of USCs when co-cultured with scaffolds. (**F**) The cell viability of USCs co-cultured with scaffolds. (**G**) The cell area of USCs co-cultured with scaffolds. (**H**) The cell viability of USCs co-cultured with scaffolds. (**I**) The number of migrated cells. * represent *p* < 0.05.

**Figure 4 pharmaceutics-16-00204-f004:**
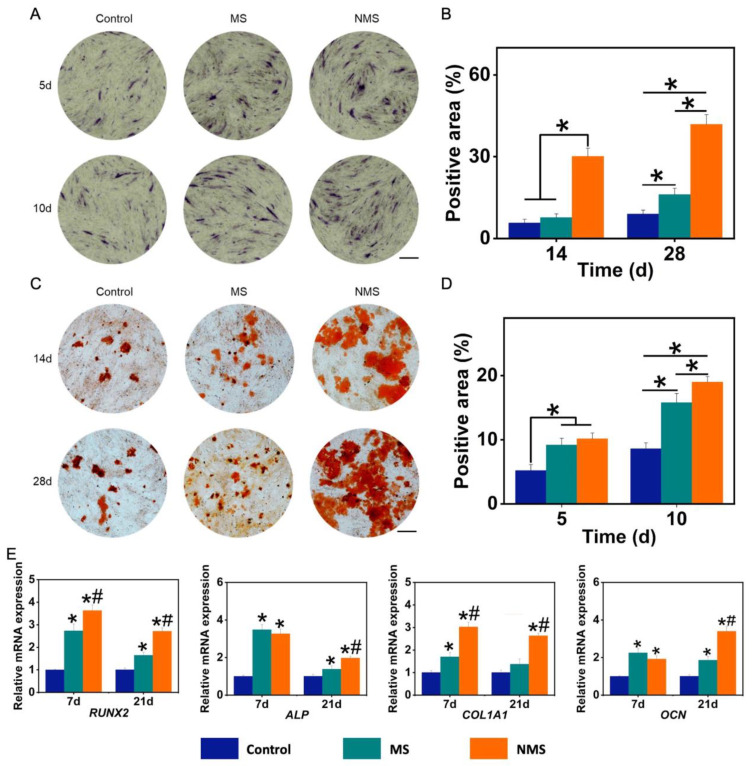
(**A**) ALP staining of USCs co-cultured with scaffolds. (**B**) The positive area of the ALP staining results. * *p* < 0.05. (**C**) ARS staining of USCs co-cultured with scaffolds. (**D**) The positive area of the ARS staining results. * *p* < 0.05. (**E**) The osteogenic-related gene expression of USCs co-cultured with scaffolds. * *p* < 0.05, compared with control, # *p* < 0.05, compared with MS.

**Figure 5 pharmaceutics-16-00204-f005:**
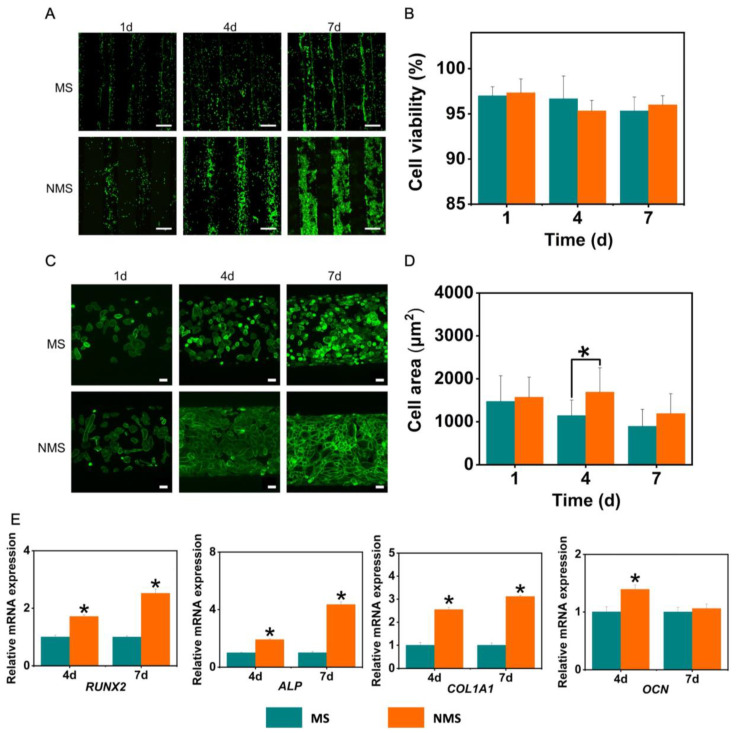
(**A**) Live/dead staining of USCs seeded onto scaffolds. (**B**) The cell viability of USCs seeded onto scaffolds. Scale bar = 500 μm. (**C**) Phalloidin staining of USCs seeded onto scaffolds. Scale bar = 50 μm. (**D**) The cell area of USCs seeded onto scaffolds. * *p* < 0.05, compared with MS. (**E**) The osteogenic-related gene expression of USCs seeded onto scaffolds. * *p* < 0.05, compared with MS.

**Figure 6 pharmaceutics-16-00204-f006:**
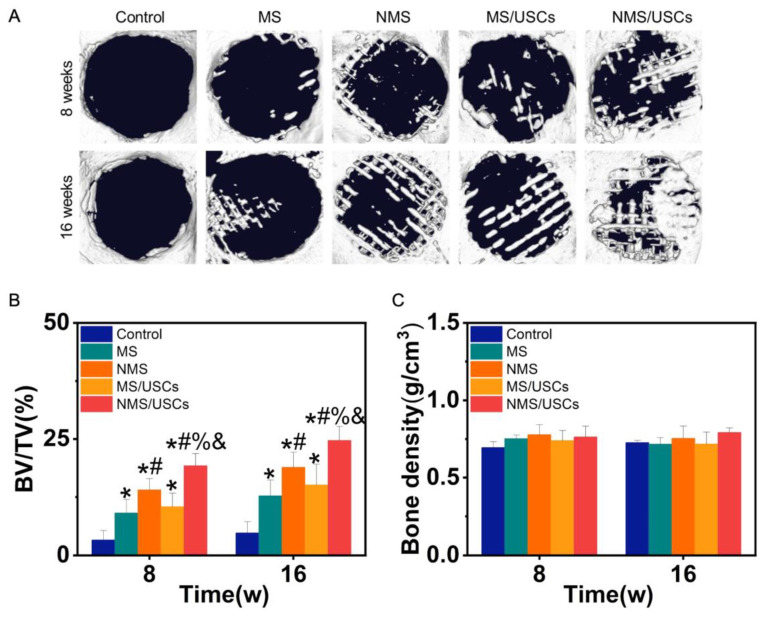
(**A**) Micro-CT results of scaffolds implanted into bone defects. (**B**) The BV/TV of each group after implantation. (**C**) The bone density of newborn bone formation in bone defects. * *p* < 0.05, compared with control, # *p* < 0.05, compared with MS, % *p* < 0.05, compared with NMS, & *p* < 0.05, compared with MS/USCs.

**Figure 7 pharmaceutics-16-00204-f007:**
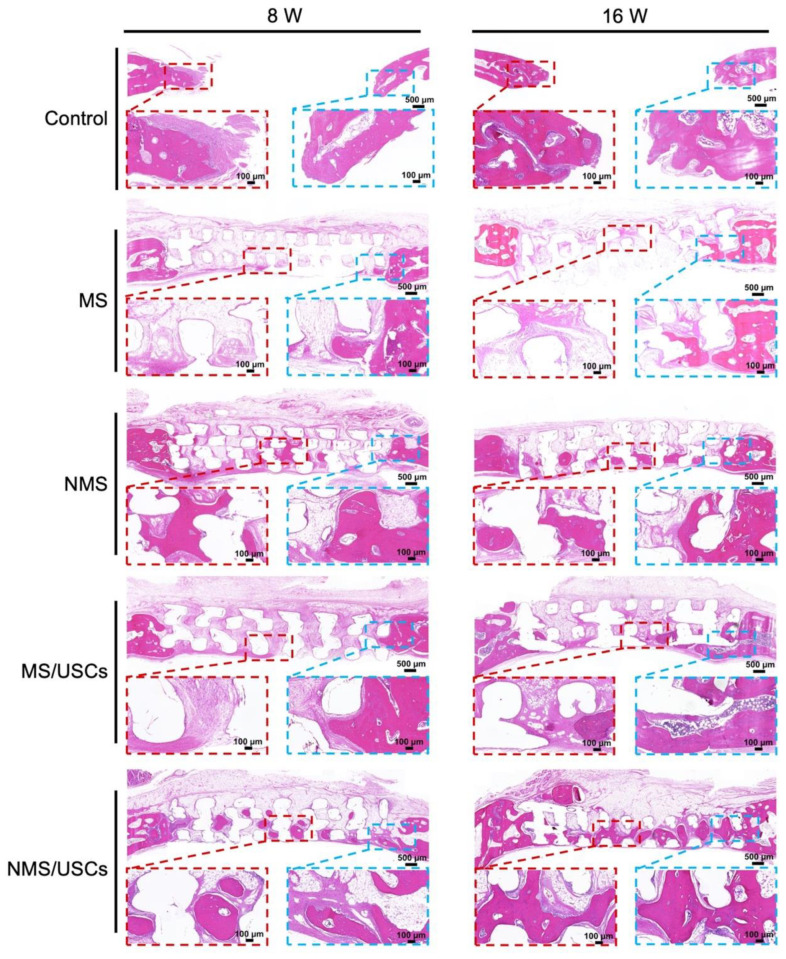
The HE staining of the bone defects after implantation.

**Figure 8 pharmaceutics-16-00204-f008:**
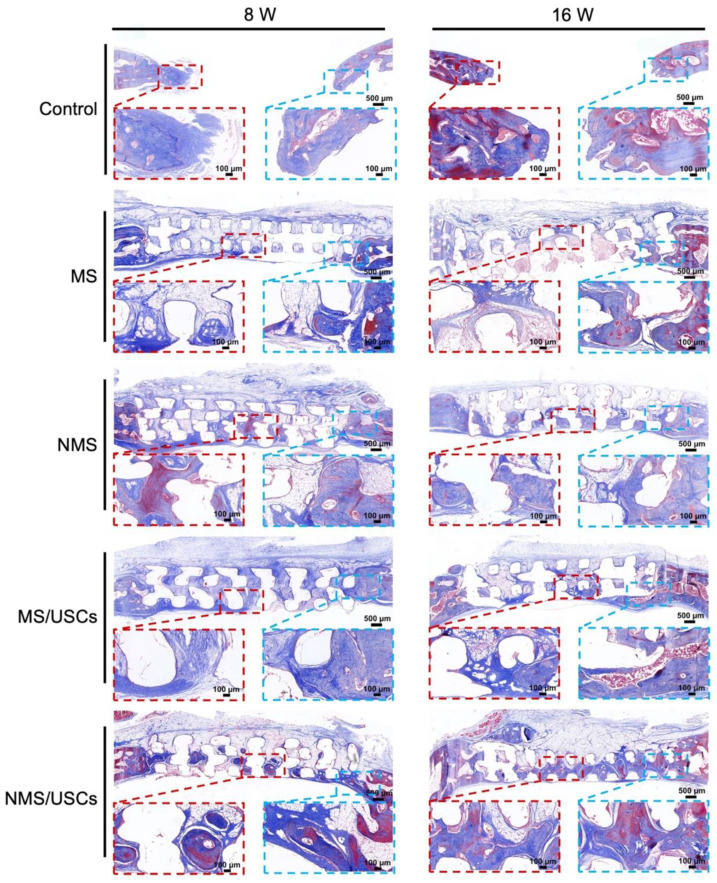
Masson staining of the bone defects after implantation.

**Figure 9 pharmaceutics-16-00204-f009:**
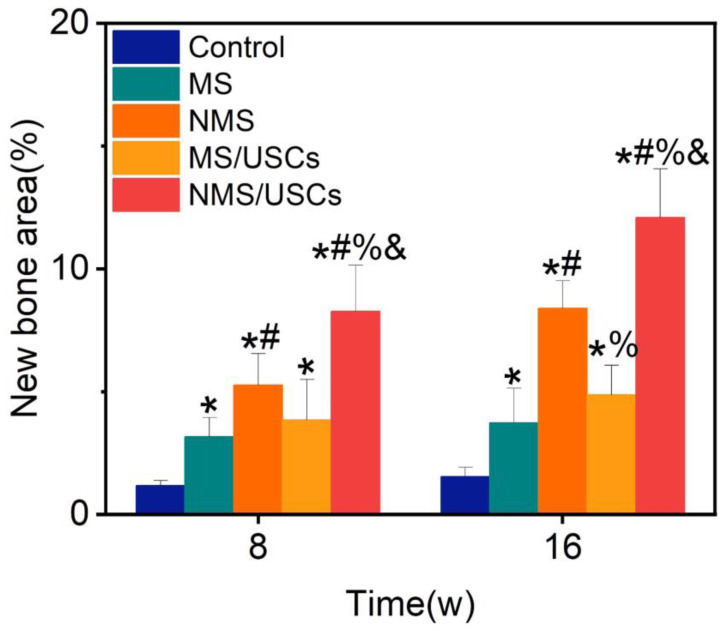
The new bone area of Masson staining results. * *p* < 0.05, compared with control, # *p* < 0.05, compared with MS, % *p* < 0.05, compared with NMS, & *p* < 0.05, compared with MS/USCs.

## Data Availability

Data are contained within the article and Appendix A.

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
