# Peer review of "Nano-Topographically Guided, Biomineralized, 3D-Printed Polycaprolactone Scaffolds with Urine-Derived Stem Cells for Promoting Bone Regeneration"

_pharmaceutics, 2024, doi:10.3390/pharmaceutics16020204_

Round 1

Reviewer 1 Report

Comments and Suggestions for Authors

This article is very interesting and can be accepted for publication

Comments

Authors made several characterizations for Nano-topographically guided biomineralized 3D-printed poly-
caprolactone, and the figures are very poor to understand your results.

Also, the statistical analysis is not clear as shown in Figures 3, 4, 5 and 6. 

Comments on the Quality of English Language

Minorr errors.

Reviewer 2 Report

Comments and Suggestions for Authors

General comments:

This paper describes PCL-based scaffolds for application of bone tissue engineering.

This study is interesting in this research field. Also, the topic is of importance.

Specific comments:

Little attention has been given to the cell experiment to investigate the cell compatibility. The appropriate sterilization method of PCL-based scaffolds (MS and NMS) is not addressed in the text.

Regarding mineralization:

The formation of MS and/or NMS surface is due to the electrostatic interaction between negatively charged layer along the BS and/or NS surface and Ca2+, which deposited calcium ions, in turn, interacted with phosphate ions (PO43-) in the SBF. The HA then grew spontaneously accompanied by consuming the calcium and phosphate ions to form apatite.

The surface charge characteristics of BS and NS should be provided in the pH range from 3 to 10, especially at pH 7.4.

After mineralization, the zeta potential of MS and NMS substrates should be presented.

In the comparison with cell membranes at physiological pH values, due to the presence of phosphatidylcholine liposomes of the cells the total negative charge is provided. In addition, the cytotoxicity depends on the charge on the substrate surface. Negatively charged material’s surface show lower unfavorable effect on the cells viability because of the negatively charged cell membrane (~ –20 mV), which plays an important role to separate the cytoplasm from the outside environment. The positively charged surfaces are more effectively adsorbed on the cell membrane as compared with the negatively charged or neutral ones. However, the positively charged surfaces cause the plasma membrane disruptions as reported by several researchers.

Regarding references: there are 17 references for 2023, 11 references for 2022, and 6 references for 2021. These references will have merit to be presented.

Reviewer 3 Report

Comments and Suggestions for Authors

The manuscript reports the analysis of the effect of the biomineralization induced on the surface of 3D printed structures made of PCL on urine-derived stem cells for bone tissue regeneration.

The work has been widely detailed, and several experiments have been performed to support the authors' hypothesis.

In the reviewer's opinion, the manuscript needs some revisions before being considered acceptable for the Pharmaceutics journal, as follows:

- The reviewer strongly suggest to consider the following articles in the introduction to discuss the influential role of roughness in the mineralization of polymeric surfaces:

Park, Jisun, et al. "Surface modification of a three-dimensional polycaprolactone scaffold by polydopamine, biomineralization, and BMP-2 immobilization for potential bone tissue applications." Colloids and Surfaces B: Biointerfaces 199 (2021): 111528.

Vannozzi, Lorenzo, et al. "Novel ultrathin films based on a blend of PEG-b-PCL and PLLA and doped with ZnO nanoparticles." ACS applied materials & interfaces 12.19 (2020): 21398-21410.

Alvarez Perez, Marco A., et al. "In vitro mineralization and bone osteogenesis in poly (ε‐caprolactone)/gelatin nanofibers." Journal of Biomedical Materials Research Part A 100.11 (2012): 3008-3019.

- As the induction of nanocues on the PCL structures is influential in determining the formation of a biomineralized coating, the authors must clarify the role of the roughness induced on the printed structure upon incubation in the nutrition solution.

- The MS structure reported in Figure 1D in the magnified image of the top view seems porous, while the starting material point is just PCL without nanotopography. Please clarify this point.

- Figure 2A reports an analysis of the nanoridges. However it is not clear how nanoridges were evaluated. Please clarify this point.

- Is the biomineralized layer formed on the MS and the NMS different in terms of composition and thickness? This discussion helps the authors clarify the role of the surface properties, and the role of the release of ions from the biomineralized layer, which influence the stem cells viability and gene expression when not seeded onto the scaffolds.

- In the conclusion paragraph, there is a sentence referred to hemolysis, while the article does not report any reference to hemolysis. Please clarify this point.

- The reviewer strongly suggests the authors modify all images to increase the clarity of the results reported and modify the captions of each figure to make them more readable, as they must clearly describe the content.

Comments on the Quality of English Language

An extensive revision of the English grammar, syntax and orthography is suggested.

Round 2

Reviewer 1 Report

Comments and Suggestions for Authors

Accept

Comments on the Quality of English Language

Minor editing of English language

Reviewer 2 Report

Comments and Suggestions for Authors

私の質問に答えていただき、それに応じて修正措置を講じていただきありがとうございます。

著者らの努力に感謝します。

回答を詳しく読みましたが、提起された問題に対処しています。

はるかに強力で出版に適しているように見えます。

Thank you for responding to my questions and taking corrective measures accordingly.

We appreciate the efforts of the authors.

After carefully reviewing the response, it appears that the issues raised have been addressed.

It seems much stronger and suitable for publication.